# How to Mitigate the Distribution Shift Problem in Robotics Control: A Robust and Adaptive Approach Based on Offline to Online Imitation Learning

## Abstract

Distribution shift in imitation learning refers to the problem that the agent cannot plan proper actions for a state that has not been visited during the training. This problem can be largely attributed to the inherently narrow state-action coverage provided by expert demonstrations over the full environment. In this paper, we propose a robust offline to adaptive online imitation learning framework that handles the distribution shift problem in a lifelong, multi-phase scheme. In the offline learning phase, we leverage supplementary demonstrations to broaden the state-action coverage of the policy by utilizing a discriminator to effectively train the policy with supplementary demonstrations, thereby enhancing the robustness of the policy to distribution shift. In the subsequent online inference phase, our framework detects the occurrence of distribution shift and conducts self-supervised imitation learning from online experiences to adapt the policy to the online environments. Through extensive evaluations in MuJoCo environments, we demonstrate that our method exhibits better robustness to distribution shift and better adaptation performance to online environments than the baseline algorithms, which indicates superior performance of our framework against the distribution shift.

## 1 Introduction

Recently, a variety of learning-based approaches such as imitation learning (IL) and reinforcement learning (RL) have achieved notable success in various robotics control tasks (Fu et al., 2024; Choi & Seo, 2025). However, these methods suffer from the distribution shift problem (Yoon et al., 2024), where the robot fails to act appropriately when encountering novel states during the online inference phase that were not present in the offline training dataset. In general, the state-action coverage of expert demonstrations spans only a narrow subset of the entire environment's state-action space, making IL especially vulnerable to distribution shift (Mehta et al., 2025; Panaganti et al., 2023). To address this, several studies have been proposed that focus on mitigating distribution shift in the offline training phase (Mehta et al., 2025; Ke et al., 2023), or in the online inference phase (Gong et al., 2024; Ho & Ermon, 2016). However, both approaches still exhibit several limitations.

For the offline phase, previous research (Mehta et al., 2025; Laskey et al., 2017a; Ke et al., 2023) aim to make the policy robust to distribution shift via dataset augmentation techniques. Laskey et al. (Laskey et al., 2017a) proposed to collect expert demonstrations in which a human intentionally encounters and recovers from various perturbation scenarios. However, this approach requires the human expert to provide demonstrations across a wide range of situations and perturbations, which is highly costly in practice. An alternative method (Ke et al., 2023) involves learning a world model from expert demonstrations and then performing data augmentation by generating a virtual dataset through rollouts of the learned model. This approach can expand the state-action coverage of the offline dataset, however, its performance is vulnerable to the modeling error (Yu et al., 2020). For the online phase, previous research (Gong et al., 2024; Ho & Ermon, 2016) focus on adapting the policy to the online environment by leveraging the agent's online experiences. Gong et al. (Gong et al., 2024) propose a lifelong imitation learning framework for navigation control that includes

self-supervised policy and distribution evaluation. However, their method does not explicitly address the distribution shift problem, making it vulnerable when encountering out-of-distribution states.

In this paper, we propose a robust offline to adaptive online imitation learning (RAIL) framework to address the aforementioned problems. During offline training, we utilize not only expert demonstrations but also supplementary demonstrations. We define supplementary demonstrations as trajectories collected from novices, suboptimal experts, or those automatically generated during agent training in simulation. All such data are inherently of unknown optimality. A key requirement is that these demonstrations can be acquired with minimal effort, in contrast to approaches that deliberately attempt to cover unexplored regions of the state space. These supplementary demonstrations provide high coverage of the full environment state space to the policy (Fig. 3).

To effectively leverage the supplementary demonstrations, we train the agent using a discriminator-based weighted behavior cloning algorithm inspired by Xu et al. (2022); Li et al. (2023). We first train a discriminator with the proposed regularization term that distinguishes between expert and supplementary demonstrations that estimate the optimality of the training samples, and then utilize it for behavior cloning. In the online phase, we build upon one important insight: from the agent's perspective, its online experience can also be regarded as suboptimal demonstrations. Based on this insight, we adopt the same learning procedure as in the offline phase for online learning. The agent computes a self-supervised learning signal from its online experience using the discriminator and updates the policy with behavior cloning. Furthermore, as online learning at every timestep could negatively impact policy generalization performance (Yoon et al., 2024), we conduct online learning only when a distribution shift is detected. When the distribution shift happens, we conduct an online update for both the discriminator and the policy from the online experience.

To evaluate RAIL framework, we design an experimental protocol that deploys the robot to the noise-injected online environment (Ke et al., 2023) to induce distribution shift. We conduct experiments on MuJoCo environments and demonstrate that our proposed method outperforms baseline algorithms in both the offline and online phases. The contributions of our paper are summarized as follows:

1. We propose a robust offline to adaptive online imitation learning framework that comprehensively addresses the distribution shift in a lifelong, multi-phase scheme.

2. We propose a regularization term for the discriminator that enables the discriminator to estimate the optimality of input data samples more accurately.

3. We present an adaptive online learning method in which the agent computes self-supervision signals from online experiences using the discriminator, and performs online learning only when a distribution shift is detected for stable adaptation.

## 2 RELATED WORKS

### 2.1 IMITATION LEARNING AGAINST DISTRIBUTION SHIFT

There are several studies that handles the distribution shift problem by making the policy robust to the distribution shift. Mehta et al. (Mehta et al., 2025) proposed to incorporate environment dynamics into the training process to improve policy robustness against distribution shift, which focus on the policy optimization procedure. The other way is to expand the dataset to encourage the policy to visit a broader range of states during training. Specifically, expert demonstrations can be collected by perturbing human operators and recording their recovery behavior, thereby providing demonstrations that are inherently more robust to distribution shift (Laskey et al., 2017a; Umut Ciftci et al., 2024). Another approach involves training a world model from the demonstrations and generating model-based virtual rollouts to augment the offline dataset (Chang et al., 2021; Ke et al., 2023). Nevertheless, these methods suffer from inherent drawbacks, such as the high cost of acquiring expert demonstrations or vulnerability to modeling errors in the learned world models.

Moreover, it is nearly impossible to include or model the infinitely many variables that may arise during the online phase within the offline dataset. In other words, distribution shift is inevitable during online inference, highlighting the necessity of online learning to address it. One of the most widely adopted approaches is RL (Schulman et al., 2015), which leverages various exploration strategies (Yoon et al., 2020) to obtain self-supervision learning signals for online learning. However, given that exploration is prohibited during inference due to operational stability or safety constraints,

adapting the policy through RL is undesirable. Furthermore, RL is fundamentally ill-suited for instant adaptation because the reward signals are often delayed, resulting in slow and unstable policy updates (Zhu et al., 2022). Upon this problem, several RL-based online imitation learning (Ho & Ermon, 2016; Yue et al., 2024) are also not desirable for the adaptation, although they can compute a self-supervision signal with a discriminator. Another line of research explores lifelong behavior cloning. Gong et al. (Gong et al., 2024) proposed computing a self-supervised behavior cloning signal for online experiences by evaluating both the policy and the distribution. However, their approach explicitly compares the online experience with the training dataset to measure novelty, which struggles to distinguish between experiences arising from distribution shift. As a result, it often fails to correctly adapt, primarily due to the difficulty in accurately computing the self-supervised learning signal.

In this paper, we propose a lifelong scheme robust offline to adaptive online imitation learning framework that addresses distribution shift by sequential mechanisms: first, by leveraging supplementary demonstrations with a proposed discriminator function during the offline phase to improve robustness to distribution shift; and second, by computing a self-supervised learning signal with the discriminator and update the policy in a stable manner to solve distribution shift during the online inference phase. In general, the target robot in which robotic intelligence performs inference are fixed. Thus, from the perspective of robot intelligence, the most practically encountered form of distribution shift is covariate shift. Accordingly, this work focuses on addressing this covariate shift.

## 2.2 LEVERAGING SUPPLEMENTARY DEMONSTRATIONS IN IMITATION LEARNING

There are previous studies that utilize supplementary demonstrations for offline imitation learning based on the problem definition that obtaining expert demonstrations is costly (Wang et al., 2023; Xu et al., 2022; Li et al., 2023). These studies train a discriminator that can distinguish between expert demonstrations and supplementary demonstrations, and perform weighted behavior cloning based on it (Xu et al., 2022). In particular, Li et al. (Li et al., 2023) applied importance weights when training the discriminator to accurately assess the importance of each demonstration. It theoretically proves that if the offline dataset covers the stationary state-action distribution of the expert policy by leveraging population-level supplementary demonstrations, policy performance can be guaranteed. Furthermore, in the context of online learning, Wang et al. (Wang et al., 2021a) and Liu et al. (Liu et al., 2025) address the issue that expert demonstrations may contain sub-optimal behaviors. Wang et al. (Wang et al., 2021a) introduces a method that addresses this by adjusting weights accordingly, while Liu et al. (Liu et al., 2025) proposed a similar approach. The core idea behind such methodologies lies in accurately estimating the level of expertise for each training sample by effectively learning the discriminator. However, a common issue is that the discriminator often fails to learn properly in the early stages of training or when the dataset is insufficient, leading to unstable learning. To address this problem, we introduce a regularization term designed to stabilize the discriminator's learning and improve overall performance. A detailed analysis of this approach is provided in Sec. 3.

## 3 WEIGHTED BEHAVIOR CLONING WITH DISCRIMINATOR

Behavior Cloning (BC) is one of the representative algorithms in imitation learning, known to enable more stable and immediate training compared to adversarial imitation learning. The generalized objective function of BC can be defined as eq. (1):

$$\min_{\pi} \mathbb{E}_{(s,a)\sim D_O} [-\omega(s,a) \log \pi(a|s)] \tag{1}$$

where $D_O$ indicates the overall training dataset. This maximizes the log likelihood between the policy's chosen action and the ground truth at a given state, where the weight $\omega(s,a)$ is calculated based on the optimality of the current state-action pair. For expert demonstration pairs, $\omega(s,a)$ is set to 1, while for the worst demonstrations, it is set to 0. For intermediate optimality, $\omega(s,a)$ takes a continuous value between 0 and 1. Thus, accurately measuring the optimality of demonstrations to set $\omega(s,a)$ is critical when training policies with BC. This optimality can be assigned either manually by humans or through discriminators. (Xu et al., 2022; Li et al., 2023) utilizes the discriminator to estimate the optimality of the offline dataset, and we build our framework upon these methods.

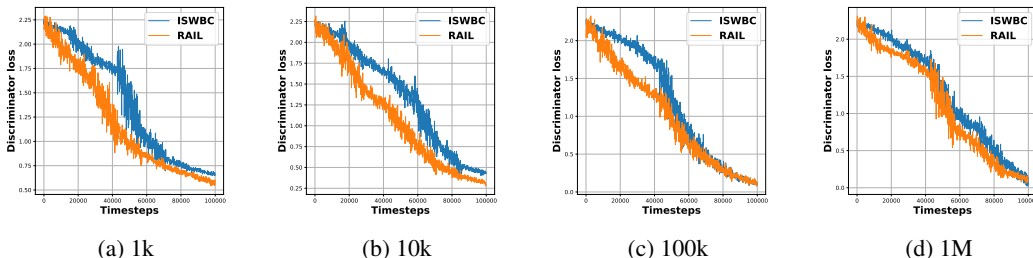

(a) 1k          (b) 10k          (c) 100k          (d) 1M

Figure 1: Discriminator evaluation loss for different sizes of expert demonstrations. The number indicated in each subcaption corresponds to the number of expert demonstrations utilized in our experiments while the number of supplementary demonstration is fixed to 1M. The x-axis of each figure denotes the training timesteps of the discriminator, while the y-axis represents the evaluation loss. Across all figures, especially in (a) - (c) that imbalance exists, it is evident that the proposed discriminator exhibits more stable and faster convergence, and also better discriminating performance compared to the baseline.

Discriminator is a strong tool to discriminate between the expert and supplementary demonstrations; however, there are several challenges in practice. When expert and supplementary demonstrations are severely imbalanced, owing to the relative ease of collecting supplementary data, the learned discriminator is prone to developing a biased decision boundary. (Arjovsky & Bottou, 2017; Mao et al., 2017). Moreover, with finite expert datasets, estimating the underlying density ratio suffers from high variance, making importance-weighted policy learning unreliable (Sugiyama et al., 2007; Thomas & Brunskill, 2016). In particular, (Li et al., 2023) assumed the population amount of the offline dataset that covers the stationary state-action distribution of the expert, which is hard to satisfy for most of the real tasks (Yue et al., 2024). This phenomenon is particularly pronounced during the early stages of discriminator training (Kiryo et al., 2017; du Plessis et al., 2016). We conduct empirical analysis about this problem, and the results are explained in Fig. 1. The core of our study lies in identifying the limitations of discriminators commonly used in online imitation learning or offline imitation learning that leverages supplementary demonstrations (Ho & Ermon, 2016; Yue et al., 2024; Li et al., 2023), and to address these limitations, we propose an improved discriminator objective by introducing a novel regularization term.

## 4 PROPOSED METHODS

### 4.1 METHOD SETTING AND OVERVIEW

We build our method on the fully observed Markov Decision Process (MDP) setting. MDP is defined by the tuple $M = (\mathcal{S}, \mathcal{A}, P, R, \mu, \gamma)$, where $\mathcal{S}$ is the state space, $\mathcal{A}$ is the action space, $T : \mathcal{S} \times \mathcal{A} \to \Delta(\mathcal{S})$ is the state transition dynamics, $R$ is the reward function, $\mu$ is the initial state distribution, and $\gamma$ is the discount factor. All policies in this paper are designed to be stochastic, based on Gaussian parameterizations as $\pi_\theta(a \mid s) = \prod_{i=1}^d \mathcal{N}(a_i; \mu_{\theta,i}(s), \sigma_{\theta,i}^2(s))$ which is paramterized by $\theta$. To train the policy, we have two types of offline datasets. The first is expert demonstrations $D_E = \{(s_E^{i,j}, a_E^{i,j})\}_{i=1}^{N_E}$, collected from an expert policy, which enable the agent to imitate expert behavior, where $N_E$ indicates the number of the trajectories in expert demonstration and $i$ indicates the index of the trajectory and $j$ indicates the index of the sample. The second is supplementary demonstrations $D_S = \{(s_S^{i,j}, a_S^{i,j})\}_{i=1}^{N_S}$, collected from a policy with unknown optimality, used to expand the state-action space coverage during offline training so that improve robustness to distribution shift. $N_S$ indicates the number of the trajectories in the supplementary demonstration. Using these demonstrations, we build our robust offline to adaptive online imitation learning (RAIL) framework that consists of two phases: the offline training and the online inference phase.

### 4.2 Offline Training Phase: Robust Imitation Learning

The main focus of the offline training phase is to effectively and stably train the policy using both expert demonstrations and optimality-unknown supplementary demonstrations. To this end, we draw inspiration from prior works (Xu et al., 2022; Li et al., 2023) and train a discriminator that can clearly distinguish between $D_E$ and $D_S$, i.e., determine the optimality of a given state-action pair provided to the training policy. Specifically, we formulate our discriminator loss function based on vanilla discriminator loss function as eq. (2) (Li et al., 2023).

$$L_{\text{disc}}^{\text{off}} = \mathbb{E}_{(s,a)\sim D_E}[-\log d(s,a)] + \mathbb{E}_{(s,a)\sim D_S}[-\frac{\widehat{d_h^S}(s,a)}{\widehat{d_h^E}(s,a)}\log(1 - d(s,a))] \qquad (2)$$

Here, $\widehat{d_h^E}(s,a)$ denotes the empirical state-action distribution of expert demonstrations, and $\widehat{d_h^S}(s,a)$ denotes that of supplementary demonstrations. It has been theoretically established that training the discriminator using the objective in eq. (2) enables binary classification between expert and supplementary demonstrations (Li et al., 2023; Ho & Ermon, 2016). Based on this, the discriminator output $d(s,a)$ and its optimal value $d^\star(s,a)$ is computed as $d(s,a) = \frac{\widehat{d_h^E}(s,a)}{\widehat{d_h^E}(s,a)+\widehat{d_h^S}(s,a)}$, $d^\star(s,a) = \frac{d_h^E(s,a)}{d_h^E(s,a)+d_h^S(s,a)}$, where the $d_h^E(s,a)$ and $d_h^E(s,a)$ are ground-truth of $\widehat{d_h^E}(s,a)$ and $\widehat{d_h^S}(s,a)$, respectively, and when trained with population-level demonstrations, $d(s,a)$ can closely approximate the ground-truth value $d^\star(s,a)$ (Li et al., 2023). Using this optimized discriminator output, the behavior cloning (BC) weight is derived as $\omega(s,a) = \frac{d^\star(s,a)}{1-d^\star(s,a)}$ (Li et al., 2023). While these findings are well-organized and have demonstrated strong performance, as analyzed in Sec. 3, the discriminator suffers from unstable training in the early stages, particularly depending on the distribution and quantity of the demonstrations.

To handle these limitations, we propose a regularization term that helps the stable and accurate learning of the discriminator. Our method aims to assist or guide the discriminator training by encouraging it to better approximate the empirical state-action distributions $\widehat{d_h^E}(s,a)$ from $D_E$ and $\widehat{d_h^S}(s,a)$ from $D_S$. Specifically, $\widehat{d_h^E}(s,a)$ corresponds to the empirical probability that a training sample $(s,a)$ originates from $D_E$, while $\widehat{d_h^S}(s,a)$ denotes the corresponding probability for $D_S$. These quantities are obtained by directly analyzing the empirical statistics of each dataset, yielding the approximations $\widehat{d_h^E}(s,a) \approx p_E(s,a)$, $\widehat{d_h^S}(s,a) \approx p_S(s,a)$, where $p_E(s,a)$ and $p_S(s,a)$ represent the estimated inclusion probabilities of $(s,a)$ in $D_E$ and $D_S$, respectively. Based on this formulation, we define the target discriminator output as $d(s,a) = \frac{\widehat{d_h^E}(s,a)}{\widehat{d_h^E}(s,a)+\widehat{d_h^S}(s,a)} \approx \frac{p_E(s,a)}{p_E(s,a)+p_S(s,a)}$, which reflects the relative likelihood that a given state-action pair originates from the expert or not. Therefore, by treating this relative ratio as a form of ground truth for training, the discriminator can be learned more stably compared to relying solely on binary discrimination. To this end, we formulate a regularization term $L_{\text{reg}}$ that guides the discriminator to this target structure, as in eq. (3).

$$L_{\text{reg}} = \mathbb{E}_{(s,a)\sim D_O}[||d(s,a) - \frac{p_E(s,a)}{p_E(s,a) + p_S(s,a)}||_2^2] \qquad (3)$$

Next, we focus on computing $p_S(s,a)$ and $p_E(s,a)$. By Bayes's rule (Murphy, 2022), these can be decomposed as $p_E(s,a) = p_E(a|s)p_E(s)$ and $p_S(s,a) = p_S(a|s)p_S(s)$, respectively. Here, $p_E(a|s)$ is approximated by the action likelihood from an expert policy $\widetilde{\pi_E}(\cdot|s)$ trained solely on $D_E$ and $p_S(a|s)$ is approximated by the action likelihood from a supplementary policy $\widetilde{\pi_S}(\cdot|s)$ trained solely on $D_S$, i.e, $p_{(\cdot)}(a|s) = \widetilde{\pi_{(\cdot)}}(a|s) = \exp(-\frac{1}{2}\sum_{i=1}^d(\log(2\pi\sigma_i^2(s)) + \frac{(a_i-\mu_i(s))^2}{\sigma_i^2(s)}))$, where $p_{(\cdot)}$ could be $p_E$ or $p_S$ with parameters for each policy. Next, we estimate $p_E(s)$ by clustering the states from $D_E$, and similarly $p_S(s)$ clustering the states from $D_S$. We utilize the Gaussian Mixture Model (GMM) approach for clustering. After fitting a GMM for each demonstration dataset, $p_E^{GMM}(s)$ for $D_E$ and $p_S^{GMM}(s)$ for $D_S$, we compute the probabilities that the training state sample is included in the fitted model, i.e., $p_{(\cdot)}^{GMM}(s) = \sum_{k=1}^K \frac{\pi_k}{(2\pi)^{d/2}|\Sigma_k|^{1/2}}\exp(-\frac{1}{2}(s -$

---

**Algorithm 1:** RAIL Framework

---

**Given:** Expert demonstrations $D_E$, Supplementary demonstrations $D_S$.
Train expert policy $\widetilde{\pi_E}$ and supplementary policy $\widetilde{\pi_S}$.
Estimate gaussian mixture model (GMM) of state $s$ : $p_E^{GMM}(s)$ of $D_E$ and $p_S^{GMM}(s)$ of $D_S$.
Train discriminator $\phi$ with $L_{\text{final}}^{\text{off}}$.
Train $\pi$ by weighted behavior cloning with eq. (1).
**while** *online inference* **do**
   **if** *distribution shift happens* **then**
      Collect online experience dataset $D_X$.
      Update $\phi$ with eq. (4).
      Update $\pi$ with eq. (1).

---

$\mu_k)^\top \Sigma_k^{-1}(s - \mu_k))$ where $p_{(\cdot)}^{GMM}(s)$ could be $p_E^{GMM}(s)$ or $p_S^{GMM}(s)$ with parameters for each fitted model. In summary, our approximated probabilities $p_E(s,a)$ and $p_S(s,a)$ could be estimated as $p_E(s,a) = \widetilde{\pi_E}(a|s)p_E^{GMM}(s)$ and $p_S(s,a) = \widetilde{\pi_S}(a|s)p_S^{GMM}(s)$.

At last, by integrating $L_{\text{reg}}$ into the discriminator objective $L_{\text{disc}}^{\text{off}}$, we stabilize the training process and improve the discriminator's ability to distinguish between expert and supplementary samples, leading to more accurate behavior cloning weight estimation. The final form of our proposed discrimination loss function is $L_{\text{final}}^{\text{off}} = L_{\text{disc}}^{\text{off}} + \lambda L_{\text{reg}}$, where $\lambda$ is the hyperparameter and it decreases as the training proceeds. Since our regularization term employs approximated values, it tends to be particularly beneficial in the initial training phase. At last, based on the output $d^\star(s,a)$ from the optimized discriminator, we formulate the behavior cloning weight $\omega(s,a) = \frac{d^\star(s,a)}{1-d^\star(s,a)}$, and perform weighted behavior cloning with eq. (1). A detailed theoretical analysis showing that $L_{\text{final}}^{\text{off}}$ enables stable learning is provided in Appendix B.1.

### 4.3 ONLINE INFERENCE PHASE: ADAPTATION VIA SELF-SUPERVISED IMITATION LEARNING

In the online phase, our core idea is that we solve the distribution shift problem via adaptation to the environment with self-supervised online learning from experience. However, indiscriminately learning from all online experiences is undesirable, as it can degrade the general policy performance (Laroche et al., 2019; Kemker et al., 2018) and also incur significant computational overhead. Therefore, we design a metric to quantify the degree of distribution shift and conduct online learning only when a distribution shift is detected. Based on the definition of distribution shift which comes from unvisited state during offline training, we define the distribution shift detection function as $\kappa(s) = \frac{p_E(s) + p_S(s)}{2}$ where $\kappa(s)$ represents the average likelihood that a given state $s$ belongs to $D_e$ or $D_s$. Instead of using the Mahalanobis distance, which is suitable for low-dimensional data, our method employs a likelihood-based estimation method that can more effectively measure distribution shift in high-dimensional robotics data (Nayal et al., 2024; Mueller & Hein, 2025). If $\kappa(s)$ falls below a predefined threshold $\kappa_{TH}$, we interpret this as an indication of distribution shift, with the severity quantified by $\kappa(s)$.

We do not immediately initiate online learning just because a single-step distribution shift is detected. While we expect the learning-based policy to handle distribution shift through its generalization capability, we also set that if distribution shift persists beyond a certain number of timesteps ($N_{ds}$), the generalization capability of the policy has failed. In such cases, we conduct online learning. We call this as method update time management (UTM). Once a distribution shift is detected, online learning is conducted using the accumulated data $D_X$ collected during the distribution shift occurrence.

At last, we describe our policy update strategy for online adaptation. When a distribution shift is detected, we perform online learning using the tuples $(s,a)$ consisting of the encountered states and the actions taken by the policy at those states. From the perspective of the policy, these tuples can be regarded as supplementary demonstrations, therefore, we adopt the same training strategy as in the offline phase. However, we incorporate the degree of distribution shift into the learning process via the $\kappa(s)$ value for adaptive learning. A high $\kappa(s)$ indicates that the corresponding state was likely visited in the offline phase, meaning that it is appropriate to sufficiently affect the discriminator that

Table 1: Result of offline training evaluation. The values in the table represent the D4RL scores obtained by performing inference with the policies trained offline by each method in an online environment with injected random Gaussian noise. Each score is averaged over 10 runs.

| Env | Noise | BC | Stable BC | CCIL | RAIL (Ours) |
|-----|-------|-----|-----------|------|-------------|
| Hopper | $\sigma = 0$ | **94.96** ($\pm$ 0.1) | 94.63 ($\pm$ 0.1) | 94.53 ($\pm$ 0.1) | 94.48 ($\pm$ 0.1) |
| | $\sigma = 0.05$ | 61.51 ($\pm$ 0.3) | 82.45 ($\pm$ 0.2) | **85.36** ($\pm$ 0.2) | 85.30 ($\pm$ 0.3) |
| | $\sigma = 0.1$ | 35.97 ($\pm$ 0.5) | 61.45 ($\pm$ 0.6) | 73.07 ($\pm$ 0.6) | **77.06** ($\pm$ 0.5) |
| | $\sigma = 0.2$ | 10.45 ($\pm$ 1.5) | 47.03 ($\pm$ 1.2) | 53.73 ($\pm$ 1.6) | **62.19** ($\pm$ 1.3) |
| Halfcheetah | $\sigma = 0$ | 92.43 ($\pm$ 0.1) | 93.30 ($\pm$ 0.1) | **93.47** ($\pm$ 0.1) | 93.28 ($\pm$ 0.1) |
| | $\sigma = 0.05$ | 48.96 ($\pm$ 0.2) | 74.05 ($\pm$ 0.3) | 84.13 ($\pm$ 0.2) | **89.27** ($\pm$ 0.3) |
| | $\sigma = 0.1$ | 24.91 ($\pm$ 0.3) | 61.02 ($\pm$ 0.5) | 70.05 ($\pm$ 0.6) | **74.07** ($\pm$ 0.6) |
| | $\sigma = 0.2$ | 5.99 ($\pm$ 1.5) | 19.48 ($\pm$ 1.7) | 42.39 ($\pm$ 1.3) | **49.57** ($\pm$ 1.9) |
| Walker2d | $\sigma = 0$ | **109.27** ($\pm$ 0.1) | 108.40 ($\pm$ 0.1) | 108.78 ($\pm$ 0.1) | 108.77 ($\pm$ 0.1) |
| | $\sigma = 0.05$ | 65.77 ($\pm$ 0.3) | 73.49 ($\pm$ 0.3) | 68.06 ($\pm$ 0.3) | **86.19** ($\pm$ 0.3) |
| | $\sigma = 0.1$ | 21.23 ($\pm$ 0.5) | 51.84 ($\pm$ 0.5) | 54.84 ($\pm$ 0.6) | **69.57** ($\pm$ 0.5) |
| | $\sigma = 0.2$ | 1.30 ($\pm$ 1.1) | 24.36 ($\pm$ 1.7) | 22.65 ($\pm$ 1.1) | **48.60** ($\pm$ 1.3) |
| Ant | $\sigma = 0$ | 92.86 ($\pm$ 0.1) | 91.79 ($\pm$ 0.1) | **93.80** ($\pm$ 0.1) | 92.57 ($\pm$ 0.1) |
| | $\sigma = 0.05$ | 39.19 ($\pm$ 0.2) | 52.45 ($\pm$ 0.3) | 59.28 ($\pm$ 0.2) | **68.20** ($\pm$ 0.2) |
| | $\sigma = 0.1$ | 21.81 ($\pm$ 0.2) | 25.56 ($\pm$ 0.4) | 29.78 ($\pm$ 0.5) | **48.68** ($\pm$ 0.6) |
| | $\sigma = 0.2$ | 5.63 ($\pm$ 1.3) | 18.77 ($\pm$ 1.1) | 21.81 ($\pm$ 1.0) | **38.68** ($\pm$ 1.0) |

was trained during the offline phase and also allow it to influence the policy (Wang et al., 2021b;a). Conversely, a low $\kappa(s)$ implies a state far from the offline dataset, and thus the update should be more conservative. Based on this reasoning, the discriminator objective used during the online phase is given by eq.(4).

$$L_{\text{disc}}^{\text{on}} = \mathbb{E}_{(s,a)\sim D_E}[-\log d(s,a)] + \mathbb{E}_{(s,a)\sim D_X}[-\kappa(s)\log(1 - d(s,a))] \qquad (4)$$

After updating the discriminator with $L_{\text{disc}}^{\text{on}}$, we perform weighted behavior cloning using the discriminator output $d(s,a)$. This approach extends behavior cloning, which was previously limited to offline imitation learning, to online imitation learning with the assistance of a discriminator. Through this method, we propose a robust offline to adaptive online imitation learning framework that can comprehensively handle the distribution shift problem during both offline training and online inference phases. The algorithm for the RAIL framework is described in Algorithm 1.

## 5 EXPERIMENTS

### 5.1 SETUP

Our research is similar to multi-task imitation learning (Zhang et al., 2023) and transfer learning (Cauderan et al., 2023) in that it continuously learns from new environments or task data. However, while these studies assume the existence of ground truth expert demonstrations for new tasks, we assume the absence of ground truth expert demonstrations for the online environment. Therefore, we conduct offline learning with an offline dataset and design an evaluation protocol by setting an online environment where a distribution shift occurs. We conduct the evaluation in four environments from MuJoCo (Hopper, Halfcheetah, Walker2d, and Ant). Expert demonstrations are constructed from the D4RL dataset (Fu et al., 2021), and supplementary demonstrations are constructed by mixing all miscellaneous demonstrations (random, medium-replay, medium, medium-expert) from the D4RL dataset. Moreover, for a unified description about the experiment results, we converted episode returns to D4RL score instead of using the raw episode returns. During online inference, random Gaussian noise is added to the state to evoke distribution shift (Laskey et al., 2017b; Mehta et al., 2025). The intensity of the noise is controlled by adjusting the covariance of the Gaussian noise $\epsilon \sim \mathcal{N}(0, \sigma^2)$. The noise intensity is set at four levels: $\sigma = 0, 0.05, 0.1, 0.2$. When $\sigma = 0$, it corresponds to the raw environment without noise, and as the value increases, the intensity of the

378
379
380
381
382
383
384
385
386
387
388
389
390
391
392
393
394
395
396
397
398
399
400
401
402
403
404
405
406
407
408
409
410
411

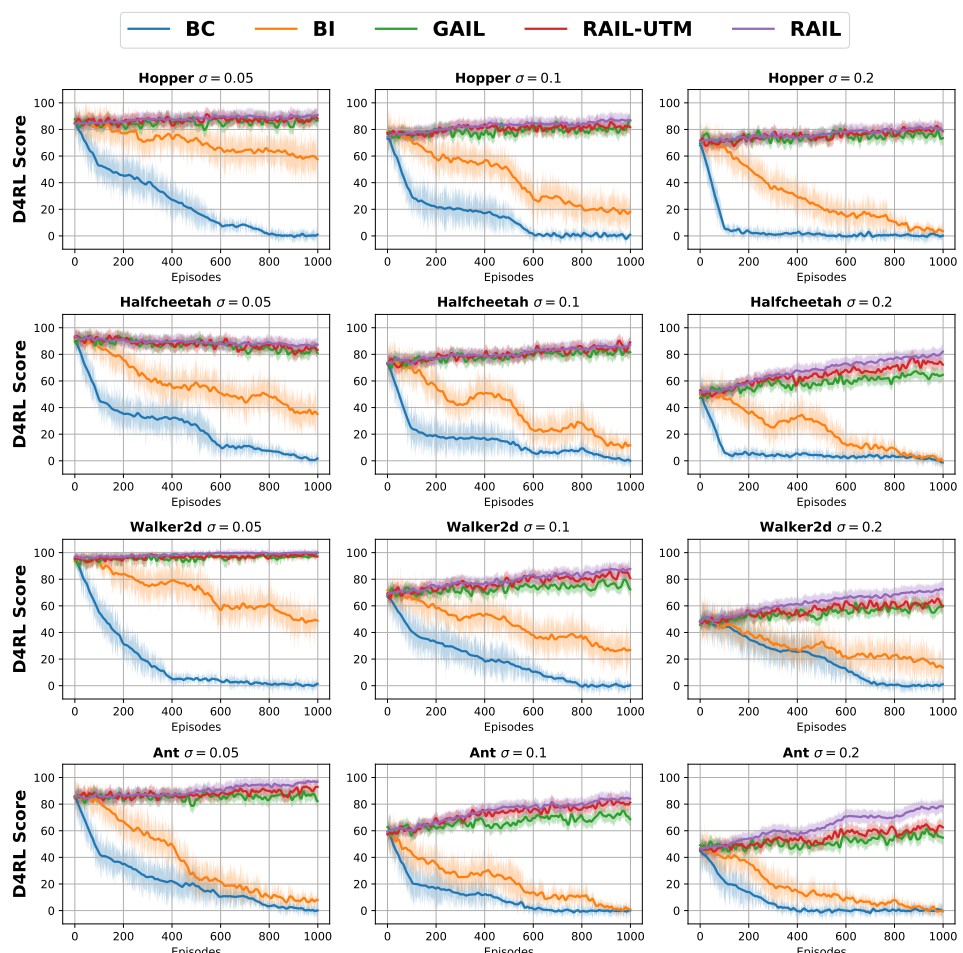

Figure 2: Evaluation for online learning performance.

412
413
414
415

noise becomes stronger. For online phase, we set $\kappa_{TH}$ to 0.4 and $N_{ds}$ to 20. We set these values as a hyperparameter and figured them out with a grid-search approach (see Appendix D.2 for details). We conduct policy training using NVIDIA RTX 3090 GPU. Through experiments, we mainly focus on answering the following questions:

416
417
418
419
420

- **Question 1 (Offline)**: Does leveraging supplementary demonstrations with the proposed discriminator function make the policy robust to the distribution shift?
- **Question 2 (Online)**: Does self-supervised online learning make the policy adapt to the current environment and solve the distribution shift?

421
422

## 5.2 RESULTS AND DISCUSSIONS

423
424
425
426
427
428
429
430
431

**Answer for Question 1.** For the offline phase evaluation to answer question 1, we adopt BC, Stable BC (Mehta et al., 2025), and Model-based BC (Ke et al., 2023) as baselines. Based on the result in Table 1, we can confirm that RAIL consistently achieves the highest single-episode return. When noise $\sigma$ is 0, the environment is identical to the offline demonstration environment, and thus all methods show similar high performance. However, as $\sigma$ increases, it becomes evident that data augmentation-based BC algorithms, Model-based BC and RAIL, are more effective than Stable BC, which focuses solely on policy optimization. Based on these results, we observe that while stabilizing policy learning is important, enhancing state coverage proves to be a more effective approach for achieving robustness to distribution shift. Among the data augmentation-based methods, the offline framework of RAIL demonstrates superior performance, verifying that leveraging real supplementary

demonstrations with our discriminator is more effective for learning a robust policy than relying on model-based virtual demonstrations. Based on these results, we can answer Q1 with a "yes."

**Answer for Question 2.** Next, for the online phase evaluation to answer question 2, we adopt BC, Beyond Imitation (BI) (Gong et al., 2024), and GAIL (Ho & Ermon, 2016) as baselines. To ensure a fair comparison, we initialized each algorithm's policy and, where applicable (i.e., GAIL and RAIL), the discriminator using the models trained during the offline phase with RAIL. Fig. 2 shows that RAIL exhibits the most significant performance improvement than other algorithms. In online phase, BC always treats online experience as expert supervision, leaving no mechanism to mitigate the compounding errors triggered by a incorrect action under distribution shift. Consequently, its performance degrades substantially. Although BI attempts to estimate self-supervision from online experience, it fundamentally assumes that the agent alone can obtain a meaningful self-supervised learning signal during the inference phase, making it vulnerable to distribution shift. In contrast, methods such as GAIL and RAIL, which compute self-supervised learning signals from online experience via the discriminator, exhibit performance improvement during online learning. This implies that they successfully address the distribution shift through adaptation from online learning. Moreover, better performance of RAIL against GAIL implies that a clear optimality estimation procedure for online experience is necessary through our regularization term. Besides, we also evaluated an ablated model, RAIL-UTM, which removes the UTM technique from RAIL and performs online learning on all online experiences without distribution shift detection. Although RAIL-UTM outperforms GAIL, it is inferior to RAIL in the overall performance. Moreover, RAIL not only outperforms RAIL-UTM, but also achieves approximately 54% reduction in training time. This result highlights the importance of the UTM technique in achieving efficient and stable adaptation.

Additionally, we further analyze GAIL and RAIL in terms of learning stability, which was introduced in Sec. 2.1. Although both methods compute self-supervised learning signals from online experience via the discriminator, GAIL updates the policy using TRPO (Schulman et al., 2015), whereas RAIL updates the policy using weighted BC. Fig. 2 demonstrates that weighted BC enables more stable policy learning during online adaptation compared to RL. More detailed experimental results are provided in Appendix C.4. Overall, RAIL consistently outperforms GAIL in terms of final performance and also guarantees more stable learning and adaptation. These results confirm that RAIL is more suitable than GAIL for achieving stable performance improvement during inference-time adaptation. Thus, we can answer Q2 with a "yes."

We further investigated the impact of supplementary demonstration coverage, with results presented in Appendix C.2. The results show that broader coverage—even from low-optimality data such as random policies—consistently outperforms using only suboptimal demonstrations with near-expert quality. This confirms that coverage, alongside optimality, is critical for policy learning and substantiates the soundness of our approach.

**Summary.** Through these extensive evaluations, we verify the effectiveness of our RAIL framework, a unified offline to online lifelong imitation learning framework. The results ensure that with training and deploying the robot with the RAIL framework, the robot policy becomes robust to distribution shift during the offline phase (to prevent distribution shift) and adapts to the online environment during the online phase (to solve distribution shift).

## 6 CONCLUSION

We propose RAIL framework that solves the distribution shift problem of imitation learning-based policy for both offline and online phase in a lifelong scheme. The key idea of our approach is to leverage supplementary demonstrations to expand the coverage of visited states during offline training. To efficiently leverage the supplementary demonstrations, we train a discriminator with regularization that estimates the optimality of training samples based on approximated probability density functions derived from expert and supplementary demonstrations. This discriminator is then used to compute BC weights, which are applied to both offline and online learning phases, enabling seamless lifelong policy training. Furthermore, by triggering online learning only when a clear distribution shift is detected, our method ensures more stable policy updates. Extensive experiments in the Mujoco environment validate the effectiveness of the RAIL framework.

REPRODUCIBILITY STATEMENT

We made extensive efforts to enhance the reproducibility of this work. The overall algorithm is summarized in Algorithm 1, and the proofs required for the method are provided in Appendix B.2. In addition, details regarding the model architecture and hyperparameters are described in Appendix D.1. Furthermore, by explicitly incorporating various details throughout the main text, we strived to maximize the reproducibility of our study.

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

## A  MOTIVATION OF LEVERAGING SUPPLEMENTARY DEMONSTRATIONS

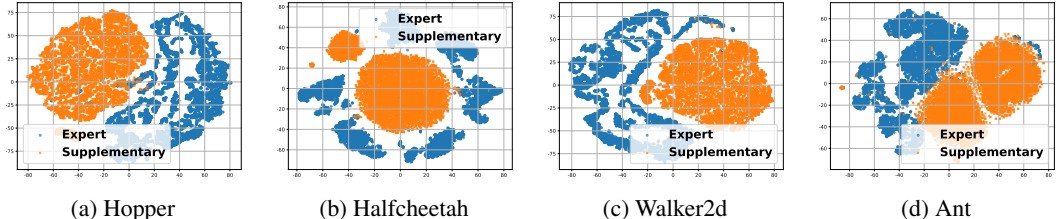

|  |  |  |  |
|---|---|---|---|
| (a) Hopper | (b) Halfcheetah | (c) Walker2d | (d) Ant |

Figure 3: t-SNE plot of the state of the expert and supplementary demonstrations of the Mujoco environment. We observe a clear discrepancy between the state distributions of expert demonstrations and supplementary demonstrations.

As shown in Fig. 3, the supplementary demonstrations and expert demonstrations cover distinct regions of the dataset. This indicates that many state–action pairs that are absent in the expert demonstrations are abundantly present in the supplementary demonstrations. Leveraging this observation, we devised a method to proactively mitigate distribution shift.

## B  THEORETICAL ANALYSIS

### B.1  POSTERIOR-REGULARIZED DISCRIMINATOR UNDER EXPERT-SUPPLEMENTARY IMBALANCE

In this subsection, we provide a theoretical analysis showing that the proposed regularized discriminator effectively mitigates the biased decision boundary caused by the imbalance between expert and supplementary demonstrations, thereby enabling stable convergence toward the optimal value $d^\star$.

**Bayes-optimal discriminator and unbiased decision boundary.** Let $p_E(s,a)$ and $p_S(s,a)$ denote the true state–action densities of the expert and supplementary demonstrations, respectively. We consider the binary classification problem of predicting whether a given $(s,a)$ originates from $p_E$ or $p_S$. Assuming equal class priors, the Bayes-optimal posterior is

$$\eta^\star(s,a) := \mathbb{P}(y = 1 \mid s,a) = \frac{p_E(s,a)}{p_E(s,a) + p_S(s,a)}. \tag{5}$$

The corresponding decision boundary is

$$\mathcal{B}^\star := \left\{ (s,a) \mid \eta^\star(s,a) = \tfrac{1}{2} \right\} = \left\{ (s,a) \mid p_E(s,a) = p_S(s,a) \right\}. \tag{6}$$

Any discriminator $d(s,a) \in (0,1)$ that aims to correctly separate the expert and supplementary regions should satisfy $d(s,a) \approx \eta^\star(s,a)$, in which case its induced boundary $\mathcal{B}_d := \{(s,a) \mid d(s,a) = \tfrac{1}{2}\}$ approximates $\mathcal{B}^\star$.

**Effect of class imbalance on the standard adversarial loss.** In practice, the discriminator is trained on empirical distributions $q_E$ and $q_S$ induced by the finite datasets $D_E$ and $D_S$. When $|D_S| \gg |D_E|$, their empirical class priors are highly imbalanced, which we express as

$$q_E(s,a) = \alpha_E p_E(s,a), \qquad q_S(s,a) = \alpha_S p_S(s,a), \qquad \alpha_S \gg \alpha_E > 0. \tag{7}$$

The standard discriminator objective is

$$L_{\text{std}}(d) = \mathbb{E}_{(s,a) \sim q_E}[-\log d(s,a)] + \mathbb{E}_{(s,a) \sim q_S}[-\log(1 - d(s,a))]. \tag{8}$$

Pointwise minimization yields the unique optimal discriminator

$$d_{\text{std}}^\star(s,a) = \frac{\alpha_E p_E(s,a)}{\alpha_E p_E(s,a) + \alpha_S p_S(s,a)} = \frac{p_E(s,a)}{p_E(s,a) + \beta\, p_S(s,a)}, \qquad \beta := \alpha_S/\alpha_E \gg 1. \tag{9}$$

Its decision boundary is

$$\mathcal{B}_{\text{std}} = \left\{ (s,a) \mid p_E(s,a) = \beta\, p_S(s,a) \right\}, \tag{10}$$

which is shifted away from $\mathcal{B}^\star$ by the imbalance factor $\beta$. Thus, the standard adversarial training objective induces a biased decision boundary whenever $\alpha_S \neq \alpha_E$.

**Posterior-regularized discriminator.** To correct this imbalance-induced distortion, we introduce the regularized objective

$$\begin{aligned} L_{\text{reg}}(d) = &\, \mathbb{E}_{(s,a) \sim q_E}[-\log d(s,a)] + \mathbb{E}_{(s,a) \sim q_S}[-\log(1 - d(s,a))] \\ &+ \lambda\, \mathbb{E}_{(s,a) \sim q_{\text{mix}}} \left[ (d(s,a) - \eta^\star(s,a))^2 \right], \end{aligned} \tag{11}$$

where $q_{\text{mix}}$ is any mixing distribution over $D_E \cup D_S$ and we notate $\frac{p_E(s,a)}{p_E(s,a)+p_S(s,a)}$ as $\eta^\star(s,a)$ for readilibty. The added term explicitly penalizes the squared deviation between the discriminator output and the Bayes-optimal posterior.

For a fixed $(s,a)$, the local objective is

$$\ell_{\text{reg}}(d; s,a) = -\alpha_E p_E(s,a) \log d - \alpha_S p_S(s,a) \log(1 - d) + \lambda \gamma(s,a)(d - \eta^\star(s,a))^2, \tag{12}$$

where $\gamma(s,a) := q_{\text{mix}}(s,a)$. The optimal discriminator $d^\star(s,a)$ satisfies the stationarity condition

$$-\frac{\alpha_E p_E(s,a)}{d^\star} + \frac{\alpha_S p_S(s,a)}{1 - d^\star} + 2\lambda \gamma(s,a)(d^\star - \eta^\star(s,a)) = 0. \tag{13}$$

Two limiting regimes follow immediately:

- **Limit $\lambda \to 0$.** Equation equation 13 reduces to the standard case, giving

$$d^\star(s,a) = d_{\text{std}}^\star(s,a) = \frac{p_E(s,a)}{p_E(s,a) + \beta p_S(s,a)}.$$

- **Limit $\lambda \to \infty$.** The quadratic penalty dominates, forcing

$$d^\star(s,a) \to \eta^\star(s,a) = \frac{p_E(s,a)}{p_E(s,a) + p_S(s,a)}.$$

Hence, the induced boundary $\mathcal{B}_{\text{reg}}$ converges to the unbiased boundary $\mathcal{B}^\star$.

For finite $\lambda > 0$, the minimizer of equation 13 lies between the imbalance-distorted optimum $d^\star_{\text{imb}}$ and the Bayes posterior $\eta^\star$. Locally approximating the cross-entropy term by a quadratic around $d^\star_{\text{imb}}(s, a)$ gives

$$d^\star(s, a) \approx \frac{w_{\text{CE}}(s, a) \, d^\star_{\text{imb}}(s, a) + w_{\text{R}}(s, a) \, \eta^\star(s, a)}{w_{\text{CE}}(s, a) + w_{\text{R}}(s, a)}, \tag{14}$$

for positive weights $w_{\text{CE}}$ and $w_{\text{R}}$ depending on $\alpha_E, \alpha_S, \lambda, \gamma(s, a)$. Thus, posterior regularization systematically pulls $d^\star$ toward the unbiased Bayes posterior, reducing the boundary distortion.

**Practical implementation via density approximation (Main proposed method).** In practice, the true densities $p_E$ and $p_S$ are unknown, so the Bayes posterior $\eta^\star$ cannot be computed exactly. Instead, we estimate the densities using a Gaussian Mixture Model (GMM) combined with a neural network encoder, obtaining consistent approximations $\widehat{p}_E, \widehat{p}_S$ and the corresponding posterior

$$\widehat{\eta}(s, a) := \frac{\widehat{p}_E(s, a)}{\widehat{p}_E(s, a) + \widehat{p}_S(s, a)}.$$

Replacing $\eta^\star$ with $\widehat{\eta}$ in equation 11 yields the empirical regularized objective. Under standard assumptions (e.g., pointwise consistency of density estimators), $\widehat{\eta}(s, a) \to \eta^\star(s, a)$, implying

$$d^\star(s, a) \xrightarrow{\lambda \to \infty} \widehat{\eta}(s, a) \xrightarrow{\text{consistency}} \eta^\star(s, a).$$

Therefore, the proposed regularizer recovers the Bayes-optimal discriminator in the limit while providing a robust finite-sample correction that mitigates the biased decision boundary induced by expert–supplementary demonstrations imbalance.

## B.2 IS THE APPROXIMATE PDF REALLY THE JOINT PDF?

Without loss of generality, we focus our exposition on the expert demonstrations. Let $D_E = \{(s_i, a_i)\}_{i=1}^N$ denote the dataset collected from expert behavior, where each tuple $(s_i, a_i)$ represents a state-action pair. Our objective is to estimate the likelihood that a training dataset $(s, a)$ originates from the expert-induced distribution. To facilitate this, we consider an approximation of the joint probability distribution as follows:

$$p_E(s, a) = p_E(a|s) \cdot p_E(s) \approx \pi_E(a|s) \cdot p_E(s) \tag{15}$$

where:

- $\pi_E(a|s)$ is the expert policy, trained from expert demonstrations $D_E$, representing the likelihood of action $a$ of the expert for a given state $s$,
- $p_E(s)$ is the marginal state distribution estimated from $D_E$, e.g., via Gaussian Mixture Model in this paper.

### VALIDITY AS A JOINT PROBABILITY DENSITY FUNCTION

We verify that our approximated probability density function $p_E(s, a) = \pi_E(a|s) \cdot p_E(s)$ satisfies the requirements of a valid joint probability density function.

**1. Non-negativity** By the definitions of conditional and marginal probability densities,

$$\pi_E(a|s) \geq 0, \quad p_E(s) \geq 0 \quad \Rightarrow \quad p_E(s, a) \geq 0 \quad \forall (s, a)$$

**2. Normalization** We must show that the integral over the full space equals 1:

$$\int_{\mathcal{S}} \int_{\mathcal{A}} p_E(s, a) \, da \, ds = \int_{\mathcal{S}} \left[ p_E(s) \int_{\mathcal{A}} \pi_E(a|s) \, da \right] ds \tag{16}$$

$$= \int_{\mathcal{S}} p_E(s) \cdot 1 \, ds = \int_{\mathcal{S}} p_E(s) \, ds = 1 \tag{17}$$

Thus, $p_E(s, a)$ is a properly normalized joint probability density function.

## B.3 DIFFERENCE FROM GAIL

The online learning procedure of the RAIL framework closely resembles that of GAIL, wherein a discriminator is trained and its output is subsequently utilized to update the policy. However, several critical distinctions differentiate RAIL from GAIL, as outlined below:

1. Unlike GAIL, RAIL introduces an additional regularization term to the discriminator objective function (eq. (3)).

2. GAIL employs TRPO (Schulman et al., 2015) to update the policy using the discriminator output, whereas RAIL adopts a weighted BC approach for policy updates that enables the online agent to stably adapt to the online environment.

3. In GAIL, action sampling is guided by online exploration to discover improved actions (Ablett et al., 2023). In contrast, RAIL forgoes exploration and instead relies solely on the self-supervised learning signal derived from the current action.

4. In summary, the differences between RAIL and GAIL are not limited to the discriminator training process; they also reflect differing algorithmic suitability depending on the task setting. GAIL is better aligned with scenarios that benefit from online exploration, whereas RAIL is more suitable for settings requiring online adaptation.

# C EXPERIMENTAL DETAILS

## C.1 D4RL DATASET

In this paper, we utilized D4RL dataset (Fu et al., 2021) (https://github.com/Farama-Foundation/D4RL) for constructing expert and supplementary demonstrations . Moreover, in order to ensure consistent and comparable evaluation across different environments, we report performance using the standardized D4RL score metric in the overall evaluation. The D4RL score is computed as

$$\text{D4RL Score} = 100 \times \frac{R_{\text{agent}} - R_{\text{random}}}{R_{\text{expert}} - R_{\text{random}}} \tag{18}$$

where $R_{\text{agent}}$ denotes the average return obtained by the evaluated policy, $R_{\text{expert}}$ is the average return of a reference expert policy, and $R_{\text{random}}$ is the average return of a random policy defined by the D4RL datasets. This normalization ensures that a score of 0 corresponds to the performance of a random policy, while a score of 100 reflects expert-level performance. The resulting metric enables direct comparison of policy effectiveness across different tasks and datasets.

## C.2 IMPACT OF SUPPLEMENTARY DEMONSTRATION COVERAGE

Our ablation results of RAIL in Table 2 and Table 3 suggest that the coverage of supplementary demonstrations plays a critical role in determining the success and robustness of adaptation, particularly under increasing noise levels. When the coverage is insufficient, the proposed method may not function as intended, highlighting a potential limitation. Nevertheless, we emphasize that the type of supplementary demonstrations employed in our framework can be obtained with minimal effort. Such data can be automatically collected during agent training in simulation, or acquired from humans, including novice users, who are relatively easy to recruit. Hence, we consider the assumption of readily available supplementary demonstrations to be reasonable, and we formulated our method accordingly. (ME: Medium-expert, M: Medium, MR: Medium-replay, R: Random)

Table 2: D4RL score with noise level 0.05.

| Supplementary demo type | Hopper | Halfcheetah | Walker2d | Ant |
|---|---|---|---|---|
| ME | 79.52 ($\pm$ 0.7) | 85.33 ($\pm$ 0.7) | 78.74 ($\pm$ 1.2) | 61.15 ($\pm$ 0.3) |
| ME + M | 82.99 ($\pm$ 0.8) | 86.19 ($\pm$ 0.6) | 81.52 ($\pm$ 0.8) | 64.08 ($\pm$ 0.7) |
| ME + M + MR | 84.30 ($\pm$ 1.2) | 88.97 ($\pm$ 0.6) | 84.97 ($\pm$ 1.8) | 66.17 ($\pm$ 0.3) |
| ME + M + MR + R | **85.30** ($\pm$ 0.3) | **89.27** ($\pm$ 0.3) | **86.19** ($\pm$ 0.3) | **68.20** ($\pm$ 0.2) |

Table 3: D4RL score with noise level 0.2.

| Supplementary demo type | Hopper | Halfcheetah | Walker2d | Ant |
|---|---|---|---|---|
| ME | 12.62 ($\pm$ 1.2) | 5.86 ($\pm$ 1.2) | 11.32 ($\pm$ 1.2) | 3.32 ($\pm$ 0.9) |
| ME + M | 44.86 ($\pm$ 1.4) | 27.77 ($\pm$ 1.0) | 34.98 ($\pm$ 1.0) | 11.28 ($\pm$ 1.1) |
| ME + M + MR | 55.47 ($\pm$ 1.4) | 33.62 ($\pm$ 1.0) | 41.01 ($\pm$ 0.8) | 22.22 ($\pm$ 1.4) |
| ME + M + MR + R | **62.19** ($\pm$ 1.3) | **49.57** ($\pm$ 1.9) | **48.60** ($\pm$ 1.3) | **38.68** ($\pm$ 1.0) |

## C.3 EFFECTIVENESS OF REGULARIZED DISCRIMINATOR

We additionally evaluated that our regularization term is helpful for discriminator training and finally guarantees better performance. For this purpose, we add our regularization term to the discriminator loss function of ISWBC (Li et al., 2023), which we got inspired by for our study. Unlike the original experiment protocol in (Li et al., 2023), we leverages 10k expert demonstrations for the realistic setting, which is hard to acquire expert demonstrations. The result is plotted in Table 4, and there is no injected noise in the online environments. That is, this experiment is purely designed to evaluate the performance of the discriminator and how effectively it contributes to improving policy learning. From the results that applying the regularization term actually increases the final performance, we demonstrate that our discriminator function is helpful for leveraging supplementary demonstrations.

Table 4: Better performance of policy with discriminator regularization.

| Algorithms | Hopper | Halfcheetah | Walker2d | Ant |
|---|---|---|---|---|
| ISWBC (Li et al., 2023) | 82.45 ($\pm$ 1.2) | 81.76 ($\pm$ 1.0) | 73.93 ($\pm$ 1.0) | 66.76 ($\pm$ 1.1) |
| RAIL (Ours) | **88.03** ($\pm$ 1.2) | **84.92** ($\pm$ 0.9) | **78.36** ($\pm$ 1.2) | **71.37** ($\pm$ 1.2) |

## C.4 LEARNING STABILITY OF RL AND IL IN ONLINE PHASE

To more specifically analyze the learning stability of RL(GAIL) and IL(RAIL), which is discussed in Sec. 2.1, we conduct an evaluation that focuses on online learning stability. To estimate learning stability, we applied an exponential moving average to the reward curve and then computed the L1 loss between the original return and the smoothed return for each episode, and averaged these values over all episodes. RAIL generally demonstrates superior results to those of GAIL, as in Table 5.

Table 5: Learning stability of GAIL and RAIL with $\sigma = 0.2$.

| Algorithm | Hopper | Halfcheetah | Walker2d | Ant |
|---|---|---|---|---|
| GAIL (Ho & Ermon, 2016) | 6.4 ($\pm$ 0.3) | 9.2 ($\pm$ 0.1) | 7.1 ($\pm$ 0.3) | **10.1** ($\pm$ 0.2) |
| RAIL | **3.5** ($\pm$ 0.2) | **5.4** ($\pm$ 0.1) | **4.9** ($\pm$ 0.3) | 14.8 ($\pm$ 0.2) |

# D IMPLEMENTATION DETAILS

## D.1 HYPERPARAMETERS

We first describe the network architecture and then the hyperparameters we used. First, for the network architecture, we build our network based on DWBC (Xu et al., 2022) (https://github.com/ryanxhr/DWBC) and ISWBC (Li et al., 2023) (https://github.com/liziniu/ISWBC). For the policy, we compose five hidden multilayer perceptron (MLP) layers with a size of $256 \times 256$. On top of this, we completed the full network architecture of the policy by incorporating an input layer and an output layer, aligned with the dimensionalities of the state and action spaces, respectively. Next, for the discriminator, we construct the discriminator network with three hidden MLP layers with a size of $256 \times 256$. The input to the network was designed to accept both the state and action simultaneously, while the output was a single float value in the range $[0, 1]$. To enhance training stability, the output was clipped to lie within the range $[0.01, 0.99]$. Related hyperparameters for

training discriminator and policy are stated in Table 6 and Table 7, respectively. $t$ in the BC weight $\lambda$ in Table 7 indicates training timesteps.

Table 6: Hyperparameters used for discriminator.

| Hyperparameter | Value |
| --- | --- |
| Learning rate | $5 \times 10^{-4}$ |
| Batch size | 64 |
| Optimizer | Adam |

Table 7: Hyperparameters used for policy.

| Hyperparameter | Value |
| --- | --- |
| Learning rate | $5 \times 10^{-4}$ |
| Batch size | 64 |
| Optimizer | Adam |
| Regularizer weight $\lambda$ | $\lambda = \begin{cases} 1, & \text{if } t \leq 10000 \\ \frac{1}{1+\log(t-9999)}, & \text{if } t > 10000 \end{cases}$ |
| $N_{ds}$ | 20 |

## D.2 PERFORMANCE SENSITIVITY TO $\kappa_{TH}$

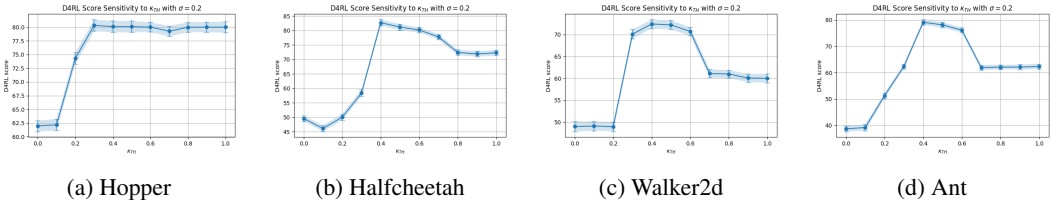

(a) Hopper     (b) Halfcheetah     (c) Walker2d     (d) Ant

Figure 4: Performance variation with respect to $\kappa_{TH}$.

We determined the optimal value of $\kappa_{TH}$ empirically through a grid-search procedure. Since the formulation of $\kappa_{TH}$ constrains it to lie within the interval $[0, 1]$, we evaluated performance at increments of 0.1, and the results are shown in Fig. 4. A smaller value of $\kappa_{TH}$ indicates that online learning is rarely triggered, whereas a larger value implies that online learning occurs at many timesteps. From these results, we observe that setting $\kappa_{TH} = 0.4$ yields the highest overall performance. This corresponds to cases where $p_E(s)$ or $p_S(s)$ lies within the range 0.8–1.0, indicating a high likelihood that the sampled state belongs to the offline dataset. Based on this observation, we set the threshold to $\kappa_{TH} = 0.4$ for all subsequent experiments.

## E THE USE OF LARGE LANGUAGE MODELS (LLMS)

In this paper, assistance from large language models (LLMs) was limited solely to the writing process, such as grammar and phrasing.

## F LIMITATIONS

A limitation of our method lies in the assumption that supplementary demonstrations visit states that are not covered by expert demonstrations, which implies no overlap. While this assumption is looser than that of prior works (Li et al., 2023), which require the expert's stationary state-action distribution to fully cover the domain, our approach may not be applicable in environments where trajectories are constrained to a limited set of states. Furthermore, accurate PDF approximation and probability-based regularization rely on the supplementary demonstrations visiting the non-expert states in a sufficiently

uniform manner. For future work, our approach may be extended by incorporating techniques for obtaining an unbiased PDF from biased demonstrations.

