# OpenReview forum: "How to Mitigate the Distribution Shift Problem in Robotics Control: A Robust and Adaptive Approach Based on Offline to Online Imitation Learning"
_ICLR.cc/2026/Conference — ICLR 2026 Conference Withdrawn Submission_

### Official Review · Reviewer_49GV · 2025-10-30

**Soundness:** 3
**Presentation:** 3
**Contribution:** 2
**Rating:** 8
**Confidence:** 4

**Summary:**

The paper introduces "RAIL", a robust offline to adaptive online imitation-learning framework for handling distribution shift in control. In the offline portion, RAIL augments narrow expert demonstrations with supplementary non‑expert/sub‑optimal trajectories and learns a discriminator‑weighted objective for BC. (Authors add a regularization term to stabalize early training) The results are evaluated on MuJoCo (Hopper, HalfCheetah, Walker2d, Ant) with Gaussian state noise.

**Strengths:**

- The paper uses the same mechanism for offline to online imitation learning as in they use a discriminator to produce weights for BC both offline which includes expert + supplementary demos as well as online which includes agent’s own experience. This is important because it's one shared pipeline (the behavior of which we can track and predict) as opposed to two unrelated stages.
- One original contribution (which is an incremental but helpful improvement) is the regularized discriminator. It's a principled stabilization of the optimality estimator used for BC weights and is especially helpful early in training.
- Another important contribution is shift detection + update‑time management. It's an intuitive way of formalizing shifts and updates occur only when low‑κ states continue and is relatively easy to implement.

**Weaknesses:**

- The paper states a fixed 3‑hour wall‑clock budget on a single RTX 3090 per environment, but sample‑efficiency and stability for TRPO‑style (GAIL) vs. weighted‑BC can be different based on environment interactions and optimizer steps. Actionable: report environment steps, updates per second, and episode counts for each baseline; include curves vs. steps.
- Although the paper does show an ablation where their regularizer is added to ISWBC’s discriminator loss, improving returns, and discriminator evaluation loss curves vs. ISWBC, this only demonstrates the regularizer helps relative to no-regularizer setup. The contribution would be stronger if there were comparisons to alternative stabilizers commonly used for density‑ratio/discriminator learning under overlap.
- The paper defines κ(s) and says online learning triggers when κ(s) falls below a threshold $κ_{TH}$ and persists for some steps (UTM), but it does not state the actual $κ_{TH}$ value, the persistence length, or provide any sensitivity sweep detection analysis.

**Questions:**

- State the exact $κ_{TH}$ value and UTM persistence length, how they were chosen, and add sensitivity sweeps showing returns and adaptation latency vs. these values.
- Sweep the regularizer weight schedule (start value, decay) and show impact on discriminator calibration and policy performance. Also compare your regularizer against at least one GAN‑style stabilizer.

---

> ### Author Response · Authors · 2025-11-28
>
> Dear reviewer 49GV
>
> First, thank you for your insightful review.
>
> We provide the following responses to the weaknesses and questions raised in the reviews.
>
> **Author response to weakness 1:**
>
> We conducted inference and online learning for both GAIL and behavior cloning over the same number of environment steps, and our measurements confirm that the computational cost associated with policy updates was identical for both methods. Additionally, the previously reported “3 hours” referred specifically to the Hopper environment; as the dimensionality increases—for instance, in Walker or Ant—the training time becomes substantially longer. To avoid potential confusion, we determined that mentioning the 3-hour figure was inappropriate and therefore removed it from the main text.
>
> **Author response to weakness 2:**
>
> We explored several alternative forms of regularization terms, but ultimately found that our proposed method yielded the most substantial performance improvement. In line with the reviewer’s suggestion, our regularization term is also designed based on a density-ratio perspective—specifically, the likelihood ratio between the priors estimated via a GMM.
>
> **Author response to weakness 3:**
>
> Through repeated experiments, we set $\kappa_{TH}$ to 0.4 and heuristically chose the UTM persistence length to be 20. Because the system is influenced by multiple interacting factors, including randomly injected Gaussian noise, compounding errors caused by distribution shift, and the inherent robustness of the policy, we determined these values empirically based on the best performance observed across multiple trials. Additional details of this procedure are provided in Appendix D.2.
>
> **Author response to Q1:**
>
> We set $\kappa_{TH}$ to 0.4 and the UTM persistence length to 20. Both values were determined heuristically through a grid search and play a central role in enabling effective self-supervised online learning. In this study, we were unable to identify a clear analytical pattern for $\kappa_{s}$, and thus we relied on heuristic tuning, which nonetheless yielded strong empirical results. Developing a principled, non-heuristic formulation for determining these values remains an important direction for future work.
>
> **Author response to Q2:**
>
> We used the regularization weight $\lambda$ as shown in Figure 7 of Appendix D.1. The key idea is that $\lambda$ remains at 1 for the first 10k steps and then gradually decays thereafter. In addition, we have included experiments comparing our method with GAIL—one of the representative GAN-style stabilizers—in Appendix C.4.

---

### Official Review · Reviewer_NPpk · 2025-10-31

**Soundness:** 3
**Presentation:** 3
**Contribution:** 3
**Rating:** 4
**Confidence:** 4

**Summary:**

The paper proposes an approach in the offline-to-online imitation setting to tackle the issue of distribution shift. In the offline phase, the method employs an additional objective to regularize the discriminator output, resulting in faster convergence. In the online phase, the method leverages the discriminator to identify distribution shifts and finetunes the policy on online data with conservatism. Experiments are conducted in Mujoco simulation with existing benchmark datasets.

Overall, the motivation is clear, and the method is sound. Experiments on simple test-time distribution shift show moderate improvements over prior methods. However, I have a few concerns that I hope the authors can help address.

**Strengths:**

* The paper is easy to follow, and the overall presentation is good.
* The online objective to adaptively update the policy and discriminator seems novel, although more ablations could be done.
* Offline-to-online imitation learning is a challenging problem, and tackling distribution shifts in BC is important.
* Results across different environments show consistent improvement over the baselines.

**Weaknesses:**

* Distinction from prior work in the offline phase is not made very clear. How does the $L^{off}_{disc}$ loss differ from (Li et al. 2023 [1])? If it is from prior work, then the authors should not claim it as their approach.
* The offline regularization seems a bit too minor for the method. I am not convinced that adding the regularization helped with performance, as the discriminator loss from Figure 2 seems to converge to a similar point, and there are no ablation studies on this component.
* The online portion of the method is interesting, but a major limitation is that with the current formulation, states that are within coverage of offline data will be used to update the discriminator/policy, but states that are OOD will be heavily discounted. This formulation could be limited when the coverage of offline data is small.
* The setting for distribution shift is too easy. The noise is only on the observations and not on actions or dynamics. Further experiments on other types of test-time distribution shift can strengthen the authors' claim.
* The authors should also consider more challenging baselines such as PWIL[2] that outperforms GAIL.

References

[1] Li et al. Imitation Learning from Imperfection: Theoretical Justifications and Algorithms (2023)

[2] Dadashi et al. Primal Wasserstein Imitation Learning (2021)

**Questions:**

* How to select the threshold $\kappa_{TH}$?
* Does the method work when the coverage of offline data is not sufficient for most online states that the agent encounters?
* What does the distribution of $\kappa(s)$ look like at different levels of distribution shifts?
* What are the assumptions/reasonings behind modeling state visitation distribution with Gaussian mixture models?
* Does the method work well in robotic manipulation domains, where the OOD issue can be essential to task performance?

---

> ### Author Response · Authors · 2025-11-28
>
> Dear reviewer NPpk
>
> First, thank you for your insightful review.
>
> We provide the following responses to the weaknesses and questions raised in the reviews.
>
> **Author response to weakness 1:**
>
> As the reviewer correctly pointed out, $L_{disc}^{off}$ is identical to the formulation used in [1]. The description in the main text was inaccurate, and we have revised the manuscript accordingly. The key point of our offline phase is that learning with our proposed regularization term ($L_{reg}$) which serves to stabilize this process. We apologize for the confusion caused by the earlier description.
>
> **Author response to weakness 2:**
>
> The revised Figure 1 now includes an ablation study showing that the severe imbalance between scarce expert demonstrations and plentiful supplementary demonstrations leads to a biased decision boundary and unstable discriminator training. Our proposed regularization mitigates this issue and stabilizes learning. Importantly, this offline regularization is only a part of our contribution; the core contribution is an offline-to-online learning framework in which a single discriminator addresses distribution shift throughout both phases.
>
> **Author response to weakness 3:**
>
> The reviewer is correct, and this indeed represents a theoretical limitation; we have explicitly stated this point in the Limitations section (Appendix F). However, from a more practical perspective, robotics is an inherently conservative domain. Deploying robot intelligence typically assumes access to a large amount of offline data and substantial pre-training before deployment, which makes our assumption appear limiting in theory but reasonably aligned with real-world practice. Nonetheless, we acknowledge that this remains a clear research limitation, and we have discussed it accordingly in the limitation section.
>
> **Author response to weakness 4:**
>
> In robotics, actions are typically normalized to a bounded range and thus distribution shifts along the action dimension (i.e., the y-axis) do not occur in practice. In contrast, the input state x can be affected by a wide variety of factors, making it the primary source of potential distributional variation. For this reason, we focus our study on covariate shift among the various forms of distribution shift. Consequently, inducing shift by injecting Gaussian noise solely into the observations constitutes a reasonable experimental protocol, and this methodology is also widely adopted in related works [1, 2].
>
> **Author response to Q1:**
>
> The threshold $\kappa_{TH}$ is treated as a hyperparameter, and we determined its value through a grid search, ultimately setting it to 0.4. The detailed grid-search procedure has been added to Appendix D.2.
>
> **Author response to Q2:**
>
> This point is addressed in our response to Weakness 3, where we have provided the relevant discussion.
>
> **Author response to Q3:**
>
> Because distribution shifts can arise from various factors—such as input noise or the degree of model training—the resulting distributions do not exhibit a specific identifiable form and instead appear noise-like, making detailed analysis challenging. Instead, we have added the grid-search procedure for $\kappa_{TH}$ in Appendix D.2, and we hope that the plotted results provide clearer insight into the behavior of $\kappa_{TH}$.
>
> **Author response to Q4:**
>
> Ultimately, the problem reduces to estimating the prior $p(o)$ over observations $o$. However, computing this probability exactly is extremely challenging given the vast space of possible observations in the real world. To address this issue, many prior works [3, 4] commonly employ clustering-based approximations of the observation space, and we adopt the same technique in our study.
>
> **Author response to Q5:**
>
> Our evaluation required many supplementary demonstrations, but existing manipulation datasets contain only a small amount of such data, and the rebuttal period was too short to collect more. Despite this limitation, our method is task-agnostic and includes no locomotion-specific design, so it naturally extends to manipulation tasks and remains effective for OOD scenarios.
>
> [1] Shaunak A Mehta, Yusuf Umut Ciftci, Balamurugan Ramachandran, Somil Bansal, and Dylan PLosey. Stable-bc: Controlling covariate shift with stable behavior cloning. IEEE Robotics and Automation Letters, 2025.
>
> [2] Michael Laskey, Jonathan Lee, Roy Fox, Anca Dragan, and Ken Goldberg. Dart: Noise injection for robust imitation learning, 2017b.
>
> [3] H. -S. Yoon, C. Kim, S. -W. Kim and S. -W. Seo, "Self-Balancing Online Dataset for Incremental Driving Intelligence," IROS 2021
>
> [4] Y. Wu, Y. He and J. Z. Huang, "Clustering Ensembles Based on Probability Density Function Estimation," CSCloud 2020

---

### Official Review · Reviewer_BVGf · 2025-11-01

**Soundness:** 3
**Presentation:** 2
**Contribution:** 2
**Rating:** 6
**Confidence:** 3

**Summary:**

This paper proposes a robust offline-to-adaptive online imitation learning framework that employs a lifetime multi-stage approach to address the distribution shift problem. In the offline learning phase, supplementary demonstrations are utilized to effectively train the policy through a discriminator, thereby expanding the policy's state-action coverage and enhancing its robustness to distribution shifts. In the subsequent online inference phase, the framework detects distribution shifts and leverages online experience for self-supervised imitation learning to adapt the policy to the online environment.

**Strengths:**

The experimental part is very solid.

**Weaknesses:**

The theoretical part is relatively lacking.

**Questions:**

Have you considered some error estimation and convergence analysis?

---

> ### Author Response · Authors · 2025-11-28
>
> Dear reviewer BVGf
>
> First, thank you for your insightful review.
>
> We provide the following responses to the weaknesses and questions raised in the reviews.
>
> **Author response to weakness :**
>
> We have added the theoretical analysis to Appendix B, including the proof of the proposed discriminator regularization and the justification for the formulation of $\kappa(s)$.
>
> **Author response to Question:**
>
> We understand convergence analysis as demonstrating that the proposed regularization term guides the loss to converge in a manner that enables the discriminator to recover an appropriate decision boundary. As mentioned in our response to the previous weakness, we have added the corresponding theoretical analysis to Appendix B.

---

### Official Review · Reviewer_nWag · 2025-11-01

**Soundness:** 2
**Presentation:** 3
**Contribution:** 2
**Rating:** 6
**Confidence:** 4

**Summary:**

The paper proposes RAIL, a two-phase imitation learning framework combining a regularized discriminator for stable offline learning with an online adaptation phase triggered by distribution-shift detection. The approach seeks to bridge the gap between offline and online IL, aiming to maintain robustness under domain shifts.

**Strengths:**

1.	The distribution shift issue in imitation learning is a well-documented challenge, and the attempt to connect offline IL with adaptive online learning is conceptually reasonable.

2.	The empirical section is reproducible and includes reasonable baselines like Stable-BC and CCIL. The code structure and methodology show attention to engineering detail.

3.	The offline-online integration is ambitious, and the inclusion of shift detection rather than naive fine-tuning demonstrates awareness of overfitting risk.

**Weaknesses:**

1.	The regularization term lacks formal motivation. The authors suggest it stabilizes the discriminator but provide neither analytical intuition (e.g., bias–variance trade-off analysis) nor empirical sensitivity results. Without such grounding, it reads as a tuning trick rather than a contribution.

2.	The κ(s) detection metric, derived from GMM density averaging, lacks statistical rigor. There is no analysis of false alarms or missed detections. Better-established OOD metrics (e.g., likelihood ratio, Mahalanobis distance) could serve as baselines but are absent.

3.	Reported improvements are small and lack statistical significance analysis. Moreover, all environments are low-dimensional continuous control benchmarks (e.g., Hopper), which do not exhibit meaningful distribution shift.

**Questions:**

1.	How does RAIL perform on visual imitation datasets (e.g., DMControl suite with pixel observations)?
2.	Would integrating uncertainty estimation (e.g., ensemble critic variance) improve adaptation reliability?
3.	Could you quantify computational overhead introduced by online adaptation?

---

> ### Comment · Reviewer_nWag · 2025-11-27
>
> The authors didn't submit any rebuttal, so I keep my score unchanged.

---

> ### Author Response · Authors · 2025-11-28
>
> Dear reviewer nWag
>
> First, thank you for your insightful review.
>
> We provide the following responses to the weaknesses and questions raised in the reviews.
>
> **Author response to weakness 1:**
>
> The ablation study on our regularization term is presented in Figure 1. ISWBC corresponds to the vanilla discriminator loss function without our regularization term, whereas RAIL denotes the discriminator loss augmented with our proposed regularization. However, as the reviewer correctly pointed out, the theoretical analysis was missing in the original submission; we have now included it in Appendix B.1.
>
> **Author response to weakness 2:**
>
> While the Mahalanobis distance mentioned by the reviewer is indeed effective for distribution-shift detection, it typically relies on the assumption of a single underlying distribution or equal covariance structures across classes. Consequently, it becomes less suitable for robotics demonstration data, where the distributional structure is often complex and the inter-class covariance differences can be substantial [1]. Motivated by these limitations, we adopt a GMM-based likelihood-ratio approach—commonly used in works such as [2]—to design our distribution-shift detection metric $\kappa(s)$.
>
> **Author response to weakness 3:**
>
> Although Hopper is a low-dimensional task, Ant represents a high-dimensional setting with 105 state dimensions. Moreover, distribution shifts affect individual state elements rather than the aggregate dimensionality per se. Therefore, we argue that our experimental setup constitutes a meaningful scenario in which non-trivial distribution shifts occur.
>
> **Author response to Q1:**
>
> Our experimental setup requires a sufficiently large amount of supplementary demonstrations, and among the available benchmarks, D4RL provides datasets that include high-quality supplementary demonstrations suitable for empirical validation. Unfortunately, D4RL does not offer benchmarks for visual imitation learning, making it infeasible to conduct visual-imitation experiments within the limited rebuttal period. However, we understood the reviewer’s question as primarily concerning the impact of increased input dimensionality. In response, we conducted additional experiments on the MuJoCo Humanoid environment, and the results are presented below.
>
> | Noise | BC | Stable BC | CCIL | RAIL |
> | --- | --- | --- | --- | --- |
> | $\sigma = 0$ | 24.13 (+-1.1) | 25.17 (+-1.1) | 27.12 (+-1.1) | 27.13 (+-1.1) |
> | $\sigma = 0.05$ | 6.67 (+-0.1) | 12.33 (+-0.0) | 15.72 (+-0.0) | 21.33 (+-0.1) |
> | $\sigma = 0.1$ | 0.01 (+-0.0) | 1.23 (+-0.0) | 2.1 (+-0.0) | 5.3 (+-0.1) |
> | $\sigma = 0.2$ | 0.01 (+-0.0) | 0.01 (+-0.0) | 1.3 (+-0.0) | 1.7 (+-0.0) |
>
> Because Humanoid is an extremely high-dimensional task, achieving strong performance with the MLP-based networks used in this paper is inherently challenging. As a result, the absolute performance is lower than in other environments; nonetheless, RAIL consistently achieves the best performance among all compared methods. In summary, RAIL remains the most effective approach even in high-dimensional state spaces, suggesting that it is likely to retain strong performance in visual imitation learning settings as well. We believe these additional results also serve as a meaningful response to Weakness 3.
>
> **Author response to Q2:**
>
> We initially considered incorporating an ensemble-based, uncertainty-aware approach during the early stages of our research. However, because our distribution-shift detection module estimates shift using deterministic probabilities, integrating an ensemble mechanism was technically nontrivial. That said, as the reviewer correctly pointed out, leveraging uncertainty-aware methods for distribution-shift detection would very likely lead to further performance improvements. Nonetheless, designing an effective integration of such methods constitutes a substantial research topic on its own, and thus we were unable to explore it within the rebuttal period. We plan to investigate this direction as part of our future work.
>
> **Author response to Q3:**
>
> We conducted an ablation study on the use of UTM during the online adaptation phase (Figure 2). The results show that our method achieves approximately 54% training-time efficiency. We have added this discussion to Section 5.2.
>
> [1] Maximilian Müller and Matthias Hein. Mahalanobis++: Improving OOD detection via feature
> normalization. In Forty-second International Conference on Machine Learning, 2025. URL
> https://openreview.net/forum?id=vutMcZl50l.
>
> [2] Nazir Nayal, Youssef Shoeb, and Fatma Güney. A likelihood ratio-based approach to segmenting unknown objects, 2024. URL https://arxiv.org/abs/2409.06424.

---

### Official Review · Reviewer_LJzn · 2025-11-03

**Soundness:** 1
**Presentation:** 3
**Contribution:** 2
**Rating:** 2
**Confidence:** 4

**Summary:**

The paper studies a unified framework that seamlessly integrates offline and online imitation learning to improve adaptability across both phases.In the offline IL phase, the authors propose a GMM-based regularization technique to stabilize and enhance the discriminator’s learning, thereby improving the policy’s robustness to distribution shift. In the online IL phase, they introduce a metric to quantify the degree of covariate shift, enabling the policy to adapt its learning behavior accordingly.

**Strengths:**

The paper makes an interesting attempt to overcome the inherent limitations of offline imitation learning, particularly its vulnerability to distribution shift and limited state-action coverage. By integrating supplementary demonstrations and discriminator-based regularization, the proposed framework aims to enhance robustness and generalization beyond standard offline IL settings. The unified design that bridges offline and online IL provides a practical perspective on how policies trained from static data can be further adapted to dynamic environments.

**Weaknesses:**

**1. Unclear problem definition**

The paper lacks a precise formulation of the problem. While it claims to address the *distribution shift problem* in imitation learning, it remains unclear *which type* of shift is considered (e.g. covariate, transition, or reward, ... ). The introduction loosely connects “lifelong” or “continual” adaptation with robustness, yet the actual experiments only inject Gaussian noise into the state to emulate distribution shift. Consequently, the notion of “robustness” and “offline-to-online adaptation” is not concretely defined or theoretically motivated. Moreover, the paper does not clearly analyze why conventional offline IL fails when applied to online adaptation (e.g., due to pessimism, limited coverage, or exploration constraints).

**2. Insufficient motivation and justification of the method**

The proposed discriminator regularization term $L_{\mathrm{reg}}$ lacks a clear theoretical basis. It heuristically enforces the discriminator output to approximate the empirical ratio of expert and supplementary densities, but the paper provides no formal analysis of how this term alleviates instability or improves density-ratio estimation. Similarly, the shift detection criterion $\kappa(s)=\frac{p_E(s)+p_S(s)}{2}$ is defined as a simple average of state likelihoods, without justification as a principled measure of covariate shift. The proposed approaches are heuristic and are only empirically validated.

**3. Limited experimental validation**

The evaluation setting is overly narrow. The only distribution shift considered is additive Gaussian noise on the state, which does not reflect more realistic transition or policy shifts in online control. The adaptation scenario is therefore limited to minor perturbations rather than genuine task or environment changes. Moreover, the comparisons omit several relevant robust IL baselines (e.g., robust IL methods). As a result, it is difficult to conclude that the proposed framework achieves general robustness or effective online adaptation.

**Questions:**

Q1. The paper repeatedly mentions *distribution shift* as the central issue, yet it remains unclear which specific distributions are shifting. Can the authors formally characterize this shift—for example, whether it refers to the discrepancy between $p_{\text{train}}(s,a)$ and $p_{\text{test}}(s,a)$, between $p_E(s)$ and $p_\pi(s)$, or between transition dynamics $P_{\text{train}}(s’|s,a)$ and $P_{\text{test}}(s’|s,a)$?

Q2. Given the definition of *distribution shift* provided in Q1, could the authors elaborate on how the proposed mechanisms (either in the offline regularization or in the online adaptation stage) specifically addresses the distribution shift?

Q3. Can the proposed method handle dynamics shifts beyond simple Gaussian noise injection? It would be helpful to clarify whether the approach can generalize to other types of dynamics changes or environment variations. If possible, the method should also be empirically evaluated and compared with relevant baselines under such dynamically changing settings.

---

> ### Author Response · Authors · 2025-11-28
>
> Dear reviewer LJzn
>
> First, thank you for your insightful review.
>
> We provide the following responses to the weaknesses and questions raised in the reviews.
>
> **Author response to weakness 1:**
>
> In typical robotics tasks, the target deployment robot is generally fixed. Consequently, distribution shift in the action space (i.e., the y-axis) is considerably less critical than shift occurring in the input state space (i.e., the x-axis), which tends to be far more detrimental. For this reason, in this paper, we focus on covariate shift, a type of distribution shift induced by changes in the input state x. To explicitly induce covariate shift in our experiments, we inject Gaussian noise into the input states.
>
> Conventional offline imitation learning assumes that online self-experience is always optimal and thus does not estimate its level of optimality of online self-experience. As a result, it cannot correct the compounding error that arises from a single action mistake. This limitation makes standard offline IL particularly vulnerable to covariate shift.
>
> We have incorporated these points into Sections 2.1 and 5.2.
>
> **Author response to weakness 2:**
>
> We have added the proof explaining how the proposed regularization-based discriminator loss function, the key contribution of this paper, improves learning stability, along with the rationale behind the design of the \kappa(s) formulation, in Appendix B and Section 4.3.
>
> **Author response to weakness 3:**
>
> We compared our method against the stable BC algorithm [1] in the offline phase and confirmed that our approach achieves superior performance. Moreover, inducing covariate shift in the online phase by injecting Gaussian noise is a widely adopted experimental protocol [1, 2]. In typical robotic systems, both the target environment and the task are fixed. Our work does not aim to address completely new tasks or environments; rather, we propose a method for resolving covariate shift that arises within an already established task and environment.
>
> The reviewer’s concern likely stems from the absence of an explicit statement in the main text emphasizing that our method specifically targets covariate shift. As mentioned in the response to Weakness 1, among various forms of distribution shift, we focus on covariate shift because it represents the most realistic challenge encountered in deployed robotic systems. We validate this choice using an experimental protocol that aligns with this setting.
>
> **Author response to Q1:**
>
> We focus on covariate shift. The answer to this question is identical to our response to Weakness 1.
>
> **Author response to Q2:**
>
> The reviewer’s point touches the central contribution of our work: a unified strategy for preventing and adapting to covariate shift. We first mitigate covariate shift by enriching expert demonstrations with diverse supplementary data to improve offline robustness. Yet covariate shift is unavoidable during online deployment, so the policy must adapt autonomously. Our method uses an offline-trained discriminator to generate a self-supervised online learning signal, allowing the policy to continually adapt to distributional changes. This offline-to-online framework thus provides a comprehensive solution to covariate shift.
>
> We have strengthened the explanation of this contribution in Section 2.1 of the revised manuscript.
>
> **Author response to Q3:**
>
> Gaussian noise is merely one of the simplest and most widely used techniques for incurring distribution shift [1, 2]. However, as the reviewer commented, we conducted other type of distribution shift by randomly varying the gravity parameter in Mujoco simulator. Specifically, a policy trained on an offline dataset collected under the standard gravity of $−9.8 m/s^2$ was evaluated in online environments where gravity was randomly sampled between $−5 m/s^2$ and $−9.8 m/s^2$.
>
> The results are plotted below and demonstrate that our method remains the most robust not only under Gaussian-noise–based perturbations but also in more general forms of distribution-shifted environments.
>
> | Environments | BC | Stable BC | CCIL | Ours |
> | --- | --- | --- | --- | --- |
> | Hopper | 17.48 ($\pm$ 1.3) | 48.88 ($\pm$ 1.5) | 23.91 ($\pm$ 1.2) | 65.77 ($\pm$ 1.4) |
> | Halfcheetah | 11.23 ($\pm$ 1.7) | 27.22 ($\pm$ 0.9) | 19.58 ($\pm$ 1.1) | 34.44 ($\pm$ 1.2) |
>
> [1] Shaunak A Mehta, Yusuf Umut Ciftci, Balamurugan Ramachandran, Somil Bansal, and Dylan P
> Losey. Stable-bc: Controlling covariate shift with stable behavior cloning. IEEE Robotics and
> Automation Letters, 2025.
>
> [2] Michael Laskey, Jonathan Lee, Roy Fox, Anca Dragan, and Ken Goldberg. Dart: Noise injection for robust imitation learning, 2017b. URL https://arxiv.org/abs/1703.09327.

---

### Author Response · Authors · 2025-11-28
**Comprehensive responses to all reviewers**

We have made every effort to address all weaknesses and questions raised by the reviewers, and we kindly ask for your understanding regarding the time required to prepare these responses. We have also uploaded the revised main text PDF with all reviewer-addressed changes. The key modifications are highlighted in blue.

---

### Author Response · Authors · 2025-12-03
**Final Comment to the Area Chair**

We would like to begin by expressing our sincere regret regarding the recent incident involving the unintended disclosure of personal information. Our submission received five reviews (maybe exceeding the typical number) and our responses were finalized later than usual because we aimed to address every weakness and question as thoroughly and carefully as possible. Given the circumstances, we understand that reviewers are no longer able to provide feedback on our author responses.

Based on our assessment, all concerns and questions raised by the reviewers have been adequately addressed through revisions to the main manuscript as well as through our detailed author responses. We respectfully submit this final comment with the hope that these efforts will be duly taken into consideration during the final decision-making process.

Thank you very much.

Sincerely,

The Authors

---

### Note · Authors · 2026-01-29

I have read and agree with the venue's withdrawal policy on behalf of myself and my co-authors.

---

### Meta-Review · Area_Chair_qRZn · 2026-01-07

**Summary:**

The paper proposes RAIL, an approach in the offline-to-online imitation setting, to tackle the issue of distribution shift. During the offline training, the method uses both expert demonstrations and supplementary demonstrations of unknown optimality and employs an additional objective to regularize the discriminator output, resulting in faster convergence. In the online phase, the method leverages the discriminator to identify distribution shifts and fine-tunes the policy on online data with conservatism. Experiments are conducted in MuJoCo with the D4RL benchmark datasets.

The main strengths of this work:
- The problem studied in this paper is well-motivated and an important challenge in imitation learning. This work showcases that using
- This work proposes a unified design that bridges offline and online imitation learning by integrating supplementary demonstrations and discriminator-based regularization
- Results on MuJoCo show improvement over the recent baselines.

On the other hand, the reviewers also raised the following major concerns and key questions:

(1) The problem definition and the scope of distribution shift are unclear (Reviewers LJzn, BVGf):

Reviewers noted that while the paper claims to study robustness and offline-to-online adaptation under distribution shift, it does not specify whether this refers to covariate, transition, or reward shift. The connection to lifelong or continual learning is loosely framed, and the experiments reduce “shift” to simple Gaussian observation noise, leaving the failure modes of offline IL in online settings (e.g., coverage, pessimism, or exploration limits) insufficiently analyzed.


(2) Insufficient theoretical grounding and heuristic design choices (Reviewers LJzn, nWag, NPpk):

Another concern from multiple reviewers is that the proposed regularization term appears heuristic, with no formal analysis explaining why it stabilizes discriminator training or improves density-ratio estimation. Similarly, the κ(s)-based shift detection criterion is not grounded in theory and is not compared against existing OOD or covariate-shift metrics.

(3) Experimental evaluation is somewhat limited (Reviewers LJzn, nWag, BVGf):

Reviewers noted that all experiments are done on low-dimensional continuous-control benchmarks with mild observation noise, without taking into account other more realistic scenarios (e.g., changes in transition dynamics, action perturbations, visual observations, or robotic manipulation).

(4) Baselines, ablations, and significance analysis (Reviewers LJzn, nWag, BVGf, NPpk):

Several reviewers pointed out that relevant robust or adversarial IL baselines (e.g., PWIL) are omitted, and that the contribution of the proposed regularization is not evaluated. Performance gains are small (Figure 2), not accompanied by statistical significance tests, and lack comparisons to alternative discriminator stabilizers commonly used in density-ratio learning.

(5) Important implementation details, sensitivity analyses, and practical costs of online adaptation are not reported (Reviewers nWag, NPpk, BVGf)

Reviewers also requested clearer exposition of several key implementation details (e.g., environment interaction counts, update frequencies, and computational overhead). Moreover, the parameter sensitivity is not specified or analyzed (e.g., κ(s) threshold, persistence length)

**Reviewer Concerns:**

During the rebuttal, the concern (2) has been well addressed, while the others appears unresolved.

(1): In the rebuttal, the authors clarified that the focus is on the covariate shift and added a brief discussion in Sections 2.1 and 5.2. This revision partially addressed the concern, but the problem formulation remains not formally stated.

(2): During the rebuttal, the authors added additional explanation on the theoretical motivation for the posterior-regularized discriminator in Appendix B.

(3): The rebuttal provided additional experiments on Humanoid, which involves higher-dimensional control than the other MuJoCo tasks. While this somewhat alleviates the concern about the generality of the method, the applicability to other more realistic scenarios and generality to other sources of covariate shifts remains unknown.

(4) and (5): The concerns about baselines, ablations, significance analysis, and sensitivity appear not addressed.

**Reviewer Scores:**

The paper received mixed initial reviews, with the reviewer scores as LJzn: 2 / nWag: 6 / BVGf: 6 / NPpk: 4 / 49GV: 8.

After the rebuttal, as many of the concerns from Reviewers LJzn, nWag, and NPpk still remain unresolved, I figure that they would tend to keep the scores unchanged if they had fully been in the discussions.

---

### Decision · Program_Chairs · 2026-01-26

Reject